# Effects of Heat Treatment on the Microstructure and Mechanical Properties of a Dual-Phase High-Entropy Alloy Fabricated via Laser Beam Power Bed Fusion

**DOI:** 10.3390/mi15040471

**Published:** 2024-03-29

**Authors:** Xiaojun Tan, Zihong Wang, Haitao Chen, Xuyun Peng, Wei Zhang, Haibing Xiao, Zhongmin Liu, Yu Hu, Liang Guo, Qingmao Zhang

**Affiliations:** 1Guangdong Provincial Key Laboratory of Nanophotonic Functional Materials and Devices, School of Information and Optoelectronic Science and Engineering, South China Normal University, Guangzhou 510006, China; tanxiaojun920@163.com (X.T.); liuzm@m.scnu.edu.cn (Z.L.); zhangqm@scnu.edu.cn (Q.Z.); 2Sino-German Intelligent Manufacturing School, Shenzhen Institute of Technology, Shenzhen 518116, China; a202415814644653@163.com (H.C.); 19970975568@163.com (X.P.); 3Intelligent Manufacturing and Equipment School, Shenzhen Institute of Information Technology, Shenzhen 518172, China; wzhang@yeah.net; 4Guangdong Provincial Key Laboratory of Industrial Ultrashort Pulse Laser Technology, Shenzhen 518055, China; 5College of Materials Science and Engineering, Chongqing University, Chongqing 400045, China; wzh_cqu@163.com; 6School of Locomotive and Vehicle, Guangzhou Railway Polytechnic, Guangzhou 510430, China; huyuscnu@163.com; 7Guangdong-Hongkong-Macao Joint Laboratory of Energy Saving and Intelligent Maintenance for Modern Transportations, Guangzhou 510430, China

**Keywords:** high-entropy alloy, laser beam power bed fusion, heat treatment, dual-phase, tailorable microstructure, mechanical property

## Abstract

To enhance the applicability of dual-phase high-entropy alloys (HEAs) like Fe32Cr33Ni29Al3Ti3, fabricated via laser beam power bed fusion (LB-PBF), a focus on improving their mechanical properties is essential. As part of this effort, heat treatment was explored. This study compares the microstructure and mechanical properties of the as-printed sample with those cooled in water after undergoing heat treatment at temperatures ranging from 1000 to 1200 °C for 1 h. Both pre- and post-treatment samples reveal a dual-phase microstructure comprising FCC and BCC phases. Although heat treatment led to a reduction in tensile and yield strength, it significantly increased ductility compared to the as-printed sample. This strength-ductility trade-off is related to changes in grain sizes with ultrafine grains enhancing strength and micron grains optimizing ductility, also influencing the content of FCC/BCC phases and dislocation density. In particular, the sample heat-treated at 1000 °C for 1 h and then water-cooled exhibited a better combination of strength and ductility, a yield strength of 790 MPa, and an elongation of 13%. This research offers innovative perspectives on crafting dual-phase HEA of Fe32Cr33Ni29Al3Ti3, allowing for tailorable microstructure and mechanical properties through a synergistic approach involving LB-PBF and heat treatment.

## 1. Introduction

The heightened interest in high-entropy alloys (HEAs) arises from their remarkable characteristics, making them well-suited for a diverse range of applications [1,2]. Conventional methods for producing HEAs present various limitations, including the need for molds or postprocessing and restricted capabilities for intricate geometries [3,4]. Additive manufacturing (AM) provides a solution by directly constructing parts layer by layer. This enables the fabrication of sizable components with complex shapes and internal characteristics [5,6]. In the realm of HEAs, different AM processes are being explored to meet modern industrial demands for intricate geometries. Particularly, LB-PBF technology distinguishes itself among these additive manufacturing methods because of its precision and rapid cooling rate [7,8]. However, the as-printed samples often exhibit significant residual stress, which is detrimental to ductility. Therefore, post-heat treatment is commonly employed to alleviate the residual stress in as-printed samples and enhance ductility [8].

Zhu et al. [9] fabricated CoCrFeNiMn samples with a single FCC phase via LB-PBF and heat-treated them at 900 °C for 1 h in an Argon atmosphere, followed by furnace cooling. After heat treatment, although the yield strength decreases, the ductility increases. Fu et al. [10] also found this phenomenon when the as-printed CoCrFeNiMn samples were heat-treated at 900 °C for 1 h and cooled in water. The microstructure, residual stress, and mechanical properties of the as-printed single FCC phase FeCoCrNi specimen and specimens heat-treated at 500–1300 °C for 2 h were compared by Lin et al. [11]. With the elevation of the heat-treating temperature, the structural transformation of the specimen occurred, transitioning from an arrangement of all columnar grains to equiaxial grains characterized by numerous heat-treating twins. Simultaneously, the dislocation network, initially formed during the solidification process under significant shrinkage strain, underwent decomposition into dislocations. This led to a reduction in residual stress, yield strength, and hardness, accompanied by an increase in ductility and impact toughness.

In the research of as-printed dual-phase HEAs, Ren et al. [12] heat-treated the as-printed samples at different conditions to study the effect of post-heat treatment on the mechanical properties of the AM EHEA (AlCoCrFeNi2.1). The conditions included 600 °C for 5 h, 660 °C for 1 h, 700 °C for 1 h, 800 °C for 1 h, 800 °C for 1 h plus 600 °C for 1 h, 900 °C for 30 min plus 600 °C for 1 h, and 1000 °C for 1 h, which were implemented in a tube furnace at a heating rate of 5 °C min^−1^ under an argon protective atmosphere followed by water quenching. When heat-treated at a temperature below 800 °C, the strength increased and the elongation decreased, above 800 °C, i.e., the opposite, when compared with the as-printed samples. Yang et al. [13] fabricated a dual-phase Ni30Co30Cr10Fe10Al18W1Mo1 HEA via LB-PBF, and the samples were heat-treated at 900 °C for 2 h followed by air cooling to alleviate the residual stress. An elongation exceeding 15% of the heat-treated samples was achieved from that of less than 1% of the as-printed samples. Vogiatzief et al. [14] heat-treated the as-printed Al0.9Cr0.9Fe2.1Ni2.1 HEA with dual-phase at 950 °C for 6 h in a furnace under an argon atmosphere, and furnace cooling was applied. After heat treatment, the elongation was improved to 20%, higher than that of the as-printed sample, by about 15%. Zhang et al. [15] investigated alterations in both the microstructure and mechanical properties of AlCoCuFeNi HEA, fabricated via LB-PBF followed by heat treatment at temperatures of 900 °C and 1000 °C for 10 h. Their results showed that heat treatment decreased the microhardness and compressive yield strength, but increased the ductility significantly as compared to the as-printed sample. In particular, the sample heat-treated at 1000 °C exhibited a better compressive fracture strength of 1600 MPa, a yield strength of 744 MPa, and a strain of 13.1%. However, these studies conducted no systematic research on heat treatment of as-printed dual-phase HEAs because too low temperatures were chosen, and the heat treatment efficiency was not taken into consideration.

In this study, we selected an as-printed dual-phase Fe32Cr33Ni29Al3Ti3 HEA as a representative specimen to investigate the impact of various heat-treated temperatures on its mechanical properties. The as-printed dual-phase Fe32Cr33Ni29Al3Ti3 HEA fabricated in our preliminary work had a high yield strength exceeding 880 MPa, but a low elongation of less than 6% [16]. Thus, post-heat treatment is necessary for as-printed samples to widen their applications. The heat-treated temperatures were set as 1000 °C, 1100 °C, or 1200 °C, and the heat-treatment time was set as 1 h according to the research of Ren [12], Zhu et al. [9], Fu et al. [10], and Zhang et al. [15]. Air cooling was attempted on the as-printed samples at 900 °C for 1 h. However, the elongation was even lower than that of the as-printed samples (see Appendix A) because of nano-precipitates via spinodal decomposition during air cooling which will be discussed in our next research. Thus, water cooling was applied to attempt to prevent precipitates [17].

## 2. Material and Methods

### 2.1. Sample Preparation

Cube samples with dimensions of 10 × 10 × 10 mm and dog-bone-shaped samples featuring a gauge length of 25 mm, width of 5 mm, and thickness of 10 mm were fabricated using a laser power of 240 W, a scanning speed of 1000 mm/s, an interlayer scanning angle of 67°, a hatch spacing of 60 μm, and a layer thickness of 30 μm. Four cube samples and two dog-bone-shaped samples were fabricated on a 304 stainless steel substrate in the same batch. Detailed information on the powder and LB-PBF strategy can be found in the previous research [16]. All the as-printed samples were cut to a thickness of 2 mm using wire EDM. After that, at least 16 cube samples and 8 dog-bone-shaped samples with a thickness of 2 mm were achieved for the subsequent heat treatment and tensile tests.

A furnace (SX_2_-5-12, LICHEN, Shanghai, China) was used for the post-heat treatment. The equipment has a heating power of 5 kW, the average heating rate is about 10 °C/min under 500 °C, and at a higher temperature above 500 °C, the heating rate will be lower due to the faster heat dissipation at the elevated temperatures. A set of heat-treated parameters was trialed for experimental optimization, as shown in Table 1. All the samples were water-cooled after being kept at their corresponding temperature for one hour. The heat-treated samples were abbreviated as 1000W-1, 1100W-1, and 1200W-1, respectively, according to their heat-treated parameters.

### 2.2. Phase and Microstructural Characterization

The phase and microstructures on the X–Y plane of both as-printed and heat-treated samples were assessed through XRD (X-ray diffraction), SEM (scanning electron microscope), and EBSD (electron backscatter diffraction) techniques (detailed information of the test equipment and the test details can be found in the previous research [16]). The X-Y plane was chosen for characterization according to the research of Fu et al. [10] in which they characterized the X-Y plane of the as-printed and annealed CoCrFeMnNi HEA samples.

Initially, the samples underwent grinding with silicon carbide paper ranging from 400 to 1500 grit, then polishing using 0.05 μm alumina polishing fluid. In SEM analysis, electrolytic etching in a 5% HCL solution for approximately 1 min was employed. EBSD analysis involved meticulous polishing for about 7 h, employing an oxide suspension with 50 nm silica particles to effectively remove the deformation layer resulting from mechanical grinding. Subsequently, the sample surfaces underwent a thorough water rinse to eliminate the nano-silica particles. The step size for EBSD measurements was set at 0.06 μm for the as-printed sample and 0.18 μm for the 1000W-1, 1100W-1, and 1200W-1 samples, considering variations in grain size. High-angle grain boundaries (HAGBs) were defined as those with a grain boundary angle higher than 15°, while low-angle grain boundaries (LAGBs) had a grain boundary angle lower than 15°. The Aztec Crystal software was utilized for the analysis of the EBSD results.

### 2.3. Tensile Test

Tensile tests were conducted on the dog-bone-shaped as-printed and heat-treated samples as in the previous research [15]. The dimensions of the dog-bone-shaped samples adhered to the specifications outlined in the ASTM standard, E1820 [18].

## 3. Results

### 3.1. Phase and Microstructure Analysis

The XRD analysis of both the as-printed and heat-treated samples is depicted in Figure 1a to verify the phase composition of FCC and BCC dual phases. A closer look at the magnified region in 2θ, ranging from 73.5° to 75.5° (Figure 1b), clearly reveals that the 2θ angles of heat-treated samples are lower than those of the original as-printed sample, with the trend indicating that the higher the heat-treated temperature, the lower the 2θ angles.

Figure 2 illustrates SEM images of the X-Y plane for both the as-printed HEA and the heat-treated samples. In Figure 2a, the microstructure of the as-printed sample reveals the BCC phase enveloped by the FCC phase. Figure 2b,c show that the BCC particles of the 1000W-1 sample are larger than those in the as-printed sample. Figure 2d shows that the BCC particles of the 1200W-1 sample are noticeably larger than those of other samples.

Additional examination of grain characteristics was performed through EBSD on the X-Y plane for both the as-printed and heat-treated samples, as shown in Figure 3. In Figure 3a, the inverse pole figure (IPF) map with grain boundaries of the as-printed HEA is presented, revealing numerous refined equiaxed grains. Figure 3b shows that the grain size of the 1000W-1 sample is quite similar to that of the as-printed sample. In Figure 3c,d, the grains of 1100W-1 and 1200W-1 samples have grown considerably when compared with those of as-printed or 1000W-1 samples, especially the 1200W-1 samples. The FCC phases of the as-printed sample show a more obvious orientation between <001> and <101>, while the heat-treated samples show a more obvious orientation of <101>. The BCC phases of the as-printed sample show a more obvious orientation <111>. In contrast, the BCC phases of the heat-treated samples show a more obvious orientation of <001>.

Figure 4 depicts the phase distribution with grain boundaries of these samples. The BCC phases of the as-printed sample show a random distribution, as shown in Figure 4a. In Figure 4b, some of the BCC phases distribute along the HAGBs of FCC phases and the other BCC phases show a random distribution among the FCC phases. This phenomenon is more severe in Figure 4c,d of the 1100W-1 and 1200W-1 samples. In Figure 4c,d, the average sizes of BCC phases of 1100W-1 or 1200W-1 samples are much larger than those in the 1000W-1 sample. Most of the BCC phases are distributed along the HAGBs and only a few small BCC phases are distributed within the FCC grains. In Figure 4b–d, many LAGBs are seen among the FCC grains.

Detailed information on the average phase sizes and their content and the content of HAGBs is shown in Table 2. $\mu_{BCC}$ and $\mu_{HAGBs}$ are the content of BCC phases and HAGBs, respectively. $d_{FCC}$ and $d_{BCC}$ are the average grain size of FCC and BCC phases, respectively. The content of BCC phases becomes less as the heat-treated temperature increases. Especially for the sample of 1200W-1, the content of BCC phases is only 13.4%, less than half of the as-printed sample. The average grain size of FCC phases first shows a little increase when comparing the 1000W-1 sample with the as-printed sample and then shows a drastic increase in the samples of 1100W-1 and 1200W-1. Especially for the sample of 1100W-1, the average grain size of FCC phases is 4.33 μm, which is more than 3.0 times larger than that of the 1000W-1 sample. The average size of the BCC phases shows a stable increase after heat-treating at 1000 °C or 1100 °C for 1 h and then cooled in water, but a drastic increase when heat-treated at 1200 °C for 1 h and then cooled in water. The average grain size of the BCC phases in the 1200W-1 sample is 2.32 μm, which is about 1.8 times larger than that of the 1100W-1 sample and 4.1 times larger than that of the as-printed sample. The content of HAGBs suffers a drastic decrease from the as-printed sample to the 1000W-1 sample and a further decrease as the heat-treated temperature increased to 1100 °C. The 1200W-1 has a similar content of HAGBs as compared to the sample of 1100W-1.

### 3.2. Mechanical Properties

Tensile tests were carried out at room temperature on as-printed and cooled HEA samples. The outcomes are illustrated in Figure 5, and the comprehensive tensile test results for the HEA samples are provided in Table 3. The as-printed sample shows a high yield strength of 913 MPa and a high tensile strength of 1239 MPa. However, the elongation is quite low, about 4.0%. After being heat-treated at 1000 °C for 1 h and then water-cooled, the yield strength decreased to 790 MPa (about a 13.5% decrease from the as-printed sample) and the tensile strength decreased to 1115 MPa. However, the elongation increased to 13.0% (about 3.0 times higher than the as-printed sample). At a higher temperature of heat treatment at 1100 °C for 1 h, the yield strength decreased by 37.5% from the as-printed sample to about 571 MPa, and the elongation increased to 20.0%. Upon further higher-temperature heat treatment at 1200 °C, the yield strength decreased to 430 MPa, and the elongation increased to 24.5%.

The fracture surface morphology is shown in Figure 6. Figure 6a shows a brittle fracture of the as-printed sample with a polyhedron rocky shape. In Figure 6b, small dimples are seen on the fracture of the 1000W-1 sample, indicating a ductile fracture. In Figure 6c, large dimples are seen on the fracture of the 1100W-1 sample. Figure 6d shows the fracture of the 1200W-1 sample, and the dimples are the largest among all these samples. The heat-treated samples all show a ductile fracture, however, the as-printed sample shows a brittle fracture.

## 4. Discussion

### 4.1. Microstructure Evolution

Figure 1b clearly shows that the 2θ angles of heat-treated samples become lower than that of the as-printed sample, which proves that the lattice distortion inside the sample was alleviated during the heat-treating process. The as-printed sample underwent rapid solidification during the LB-PBF process layer by layer, and the residual stress was very large inside the sample. Thus, the lattice distortion was serious inside the sample [13]. However, after being heated at a high temperature and kept for a certain amount of time, the dislocation network, originating from the solidification process subjected to substantial shrinkage strain, underwent decomposition into individual dislocations, and the lattice distortion inside the sample was alleviated [11].

The FCC phases of the as-printed sample show a more obvious orientation between <001> and <101>, which was also found by Fu et al. [10] in the as-printed CoCrFeMnNi HEA. After heat treatment, the FCC phases show a more obvious orientation of <101>, which means that the FCC phases grow easily in the <101> direction during heat treatment [15]. The BCC phases of the as-printed sample show a more obvious orientation <111>, which may be the result of large residual stress after LB-PBF. In contrast, the BCC phases of the heat-treated samples show a more obvious orientation of <001>. <001> is quite frequently observed for BCC metals after additive manufacturing and the BCC phases grow easily in the <001> direction during heat treatment [14].

The content of HAGBs suffers a drastic decrease from the as-printed sample to the heat-treated samples. This observation aligns well with the findings obtained from XRD analysis. HAGBs often have higher energy than LAGBs. A higher content of HAGBs means a severe lattice distortion [19,20]. A further decrease in the content of HAGBs happens when the heat-treating temperature increases to 1100 °C, which means that a higher heat-treating temperature of 1100 °C is helpful to release the residual stress [11]. The 1200W-1 has a similar content of HAGBs as compared to the sample of 1100W-1, which means that the residual stress has almost been released over a temperature of 1100 °C, so the content of HAGBs changes little. The heat-treating method has been used to alleviate the residual stress of as-printed samples in many reports [11,13].

The BCC phases grew larger after heat treatment as shown in Figure 4 and Table 2. This is because when a material is heated to such high temperatures, the atoms within the grains diffuse and rearrange themselves to minimize their energy. This can result in the coalescence of smaller grains into larger ones, which would make the BCC phases appear larger. Ostwald [21] found the Ostwald ripening phenomenon, which shows that smaller grains dissolve, and their material reprecipitates on the surface of larger grains. This can lead to the growth of larger BCC grains and a corresponding reduction in the number of smaller ones. The average grain size of BCC phases shows a stable increase after being heat-treated at 1000 °C or 1100 °C for 1 h and then cooled in water, but a drastic increase when heat-treated at 1200 °C for 1 h and then cooled in water, which means that a higher heat-treated temperature is good for the growth of the BCC grain size.

The size of the FCC phase of the 1000W-1 sample is similar to that of the as-printed sample, as shown in Figure 4 and Table 2. The BCC phases with a small average size of 0.82 μm, which are distributed along the FCC grain boundaries, can pin grain boundaries, thus inhibiting the growth of FCC phases during heat treatment. However, after undergoing heat-treating at 1100 °C for 1 h, the FCC phases become more than 3.0 times larger than those of the 1000W-1 sample. The BCC phases of the 1100W-1 sample have a large average size of 1.29 μm and the number of BCC phases decrease considerably when compared with as-printed or 1000W-1 samples, the pinning effect is weakened, and the FCC phases of the 1100W-1 sample grow to an average size of 4.33 μm. After heat-treating at a higher heat-treating temperature of 1200 °C, the FCC phases grow to 5.8 μm. Oikawa et al. [22] found the pinning effects of second-phase particles with an average size of 0.28 μm influencing the grain growth of ferrite in a Fe-0.1C alloy enriched with 5 ppm of boron (B). Li et al. [23] introduced a phase field model to explore how particle pinning influences the migration of grain boundaries in materials with stored energy variations along the grain boundaries. The outcomes indicated that reducing the particle size and increasing the particle area fraction could amplify the pinning effect. 

The content of FCC phases increases and the content of BCC phases decreases with an increase in heat-treating temperature. At a certain temperature, the BCC phase will convert to the FCC phase, and the heat during heat treatment makes this happen. Panda et al. [24] heat-treated AlCoCrFeNi (prepared via arc melting, BCC + B2 structure) in the temperature range starting from 1073 K to 1373 K up to 10 h. The results showed that the formation of the FCC phase was up to 30~35%, indicating that a long time in a high-temperature range will lead to the BCC phase transforming to the FCC phase.

During heat treatment, nucleation and growth of new phases can occur. HAGBs of the FCC phases provide sites for nucleation of the BCC phases because they often have higher energy. The large BCC phases may nucleate along these boundaries due to favorable thermodynamic conditions and then grow as heat treatment continues. Zhang et al. [25] suggested a thermomechanical treatment method capable of adjusting the lamellar structure of duplex stainless steel to form an equiaxed matrix of austenite and ferrite. Their observations revealed a preference for alpha martensite nucleation at the large-angle grain boundaries (HAGBs) between austenite grains. The small BCC phases that remained among the FCC phases represent that the phase transformation was less severe in these regions because these regions consist of LAGBs with lower energy than that of HAGBs.

### 4.2. Strengthening Mechanism

The tailorable mechanical properties of Fe32Cr33Ni29Al3Ti3 fabricated via LB-PBF before and after heat treatment can be explained by the following:

(1) The tailorable mechanical strength of Fe32Cr33Ni29Al3Ti3 fabricated via LB-PBF before and after heat treatment is attributed to the tailorable grain size and the influence of dislocation density.

To investigate this trend, the primary mechanisms contributing to strengthening were examined as a combination of dislocation hardening and grain refinement [26]. This can be succinctly expressed in the following equation:(1)$${\Delta \sigma }_{YS}=\sigma_{0}+\Delta \sigma_{gb}+\Delta \sigma_{disl}$$
where $\sigma_{YS}$ represents the yield strength of these samples, and $\Delta \sigma_{gb}$ and $\Delta \sigma_{disl}$ denote the strengthening contributions from grain boundaries and dislocations, respectively.

The lattice frictional strength of the present alloy, denoted as $\sigma_{0}$, can be regarded as 267 MPa, according to (CoCrNi)94Al3Ti3 HEA reported by Zhao et al. [27] which has almost the same elemental composition as Fe32Cr33Ni29Al3Ti3 (Fe was replaced by Co. Fe and Co are both iron-based elements, sharing similar physicochemical properties). The $\Delta \sigma_{gb}$ can be calculated by the optimized Hall-Petch equation according to our previous research [16]:(2)$$\begin{array}{cc} \Delta \sigma_{gb}=\sigma_{gb}- & \sigma_{0}=\sigma_{0}+K\times d^{-\frac{1}{2}}-\sigma_{0}=K\times d^{-\frac{1}{2}}\\ & =\mu_{FCC}\times k_{\mathrm{FCC}}\times {d_{\mathrm{FCC}}}^{-\frac{1}{2}}+\mu_{BCC}\times k_{\mathrm{BCC}}\times {d_{\mathrm{BCC}}}^{-\frac{1}{2}} \end{array}$$
where d represents the average grain size, and K is a grain boundary strengthening coefficient. $\mu_{FCC}$ is the content of the FCC phase of each sample; $\mu_{BCC}$ is the content of the BCC phase of each sample, $k_{\mathrm{FCC}}$ and $k_{\mathrm{BCC}}$ are the strengthening coefficients of the FCC-FCC interface and the FCC-BCC interface, respectively; the coefficients $k_{\mathrm{FCC}}$ and $k_{\mathrm{BCC}}$ were considered as 275 MPa∙μm^1/2^ and 574 MPa∙μm^1/2^, respectively [28,29]. The value of $k_{\mathrm{BCC}}$ is higher than that of $k_{\mathrm{FCC}}$ because the BCC phase displays greater resistance to dislocations compared to the FCC phase [30]. $d_{\mathrm{FCC}}$ and $d_{\mathrm{BCC}}$ represent the average grain sizes of the FCC and BCC phases in each sample, respectively. The data in Table 2 were input into Equation (2), and $\Delta \sigma_{gb}$ was calculated to be 396 MPa, 332 MPa, 205 MPa, and 149 MPa for as-printed, 1000W-1, 1100W-1 and 1200W-1 samples, respectively.

In LB-PBF-processed materials, the high dislocation density often plays a crucial role in achieving high strength levels [31]. The dislocation hardening effect can be described by the Taylor equation, which is expressed as follows [12,32]:(3)$$\Delta \sigma_{disl}=M\alpha Gb\rho^{1/2}=\mu_{FCC}M_{FCC}\alpha_{FCC}G_{FCC}b_{FCC}\rho_{FCC}^{1/2}+\mu_{BCC}M_{BCC}\alpha_{BCC}G_{BCC}b_{BCC}\rho_{BCC}^{1/2}$$
where M is the Taylor factor (3.09 and 2.71 for FCC and BCC materials, respectively), α is a constant (0.20 and 0.24 for FCC and BCC materials, respectively), G is the shear modulus (81 and 57 GPa for FCC and BCC materials, respectively), b is the Burgers vector (0.254 and 0.248 nm for FCC and BCC materials, respectively), and ρ is the dislocation density. The values of M, α, G, b were achieved from the research of Ren et al. [12]. The dislocation density was calculated using the Williamson-Hall method based on the XRD pattern (Figure 1) [33,34]. The measured values of ρ are shown in Table 4. These data were input into Equation (3), and $\Delta \sigma_{disl}$ could be estimated to be 179 MPa, 133 MPa, 132 MPa, and 125 MPa for as-printed, 1000W-1, 1100W-1, and 1200W-1 samples, respectively, as shown in Table 4.

Upon substituting each strengthening contribution into Equation (1), the predicted yield strengths are approximately 842 MPa, 732 MPa, 604 MPa, and 541 MPa for the as-printed, 1000W-1, 1100W-1, and 1200W-1 samples, respectively. These values demonstrate a satisfactory agreement with our experimental data, as depicted in Figure 7. The main cause of error between predicted yield strength and actual yield strength is that K (grain boundary strengthening coefficient) may differ in these samples. Huang et al. [35] found that the Hall-Petch slopes (K values) slightly increased with offset strain. In this study, the strain shows a decrement with the increase in heat-treating temperature as the dislocation density becomes lower and lower. The value of K was supposed to become smaller and smaller with the increase in heat-treating temperature. But in the calculation by Equation (2), it was set as the same. So the actual yield strength was higher than the predicted yield strength in as-printed and 1000W-1 samples, and lower than the predicted yield strength in 1100W-1 and 1200W-1 samples.

(2) The tailorable ductility of Fe32Cr33Ni29Al3Ti3 fabricated via LB-PBF before and after heat treatment is attributed to the tailorable grain size.

The average grain sizes of as-printed, 1000W-1, 1100W-1, and 1200W-1 samples are 0.69, 1.00, 1.94, and 3.70 μm, respectively. The grains of the as-printed sample are ultra-fine grains, with an average grain size of less than 1 μm. And the grains of other samples are micron grains. The relationship between elongation and the average grain sizes is shown in Figure 8.

The relationship between elongation and the average grain size fits well with the influence of grain size on elongation studied by other researchers [36,37]. The elongation increases with the average grain size increase. The samples with micron grains have an elongation higher than 10%. The as-printed sample shows a sharp decrease in elongation from the 1000W-1 sample because of its ultra-fine grains. When the grain size is less than 1.00 μm, the elongation decreases sharply [36].

The content of FCC and BCC phases is another factor that influenced the elongation of these samples. The higher content of the FCC phase is good for the ductility of a dual-phase sample [15]. As the heat-treating temperature increases, the content of the FCC phase also increases, contributing positively to ductility.

The 1000W-1 sample exhibited a better combination of yield strength and ductility, a yield strength of 790 MPa (only a 13.5% decrease from the as-printed sample), and an elongation of 13% (higher than 10%, about 3.0 times higher than the as-printed sample). The 1100W-1 and 1200W-1 samples exhibited a worse combination of yield strength and ductility because they show large decreases (more than 37%) in the yield strength when compared with the as-printed sample, though they have elongation higher than 20%. However, these tailorable mechanical properties are all valuable, and their applications depend on the need.

## 5. Conclusions

This study investigated different heat treatment parameters for the mechanical performance optimization of a HEA devoid of cobalt, characterized by a composition comprising Fe32Cr33Ni29Al3Ti3 fabricated via laser beam power bed fusion (LB-PBF). Heat treatment decreased the yield strength, but increased the ductility significantly when compared with the as-printed sample. The tailorable mechanical strength of Fe32Cr33Ni29Al3Ti3 fabricated via LB-PBF before and after heat treatment is attributed to the tailorable grain size and the influence of dislocation density (${\Delta \sigma }_{YS}=\sigma_{0}+\Delta \sigma_{gb}+\Delta \sigma_{disl}$). The as-printed sample with ultra-fine grains, higher content of BCC phases, and higher dislocation density showed higher strength and lower elongation, while heat-treated samples with micron grains, lower content of BCC phases, and lower dislocation density showed lower strength and higher elongation. In particular, the sample heat-treated at 1000 °C for 1 h and water-cooled exhibited a better combination of yield strength and ductility, a yield strength of 790 MPa, and an elongation of 13%. This study may have particular value in the post-heat treatment of dual-phase HEAs fabricated via LB-PBF.

## Figures and Tables

**Figure 1 micromachines-15-00471-f001:**
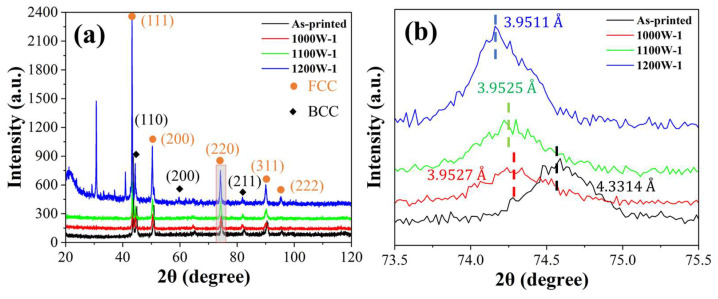
(**a**) XRD patterns; (**b**) partial XRD patterns of the as-printed and heat-treated HEA samples with different heat-treated parameters.

**Figure 2 micromachines-15-00471-f002:**
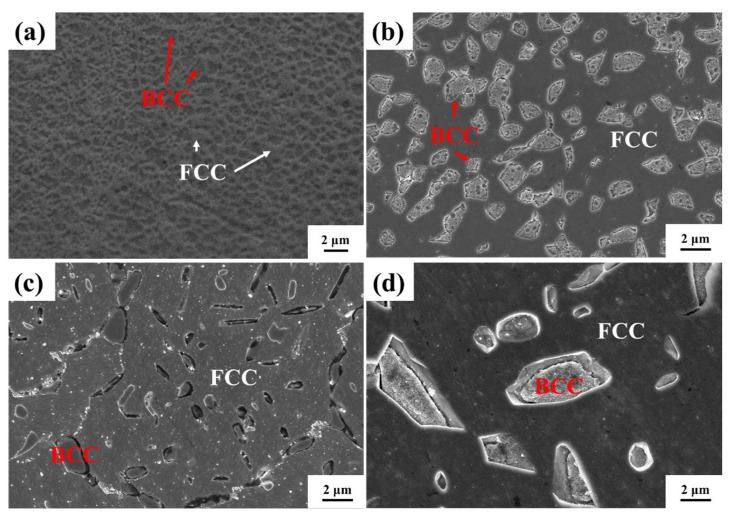
SEM microstructures of the X-Y plane of the as-printed HEA (**a**) and heat-treated samples: 1000W-1 (**b**); 1100W-1 (**c**); 1200W-1 (**d**).

**Figure 3 micromachines-15-00471-f003:**
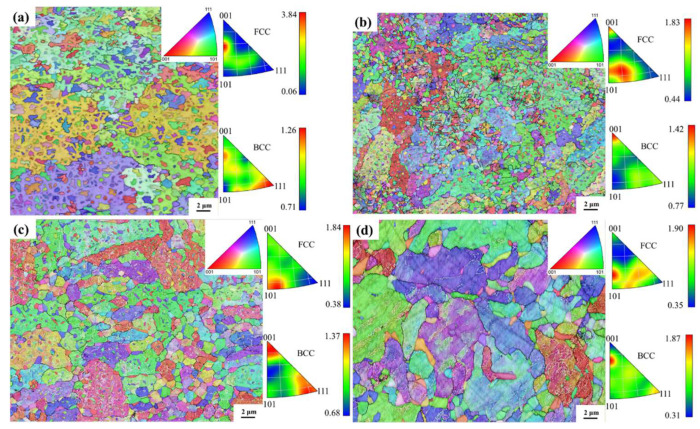
IPF map with grain boundary of the as-printed HEA (**a**) and heat-treated samples: 1000W-1 (**b**); 1100W-1 (**c**); 1200W-1 (**d**).

**Figure 4 micromachines-15-00471-f004:**
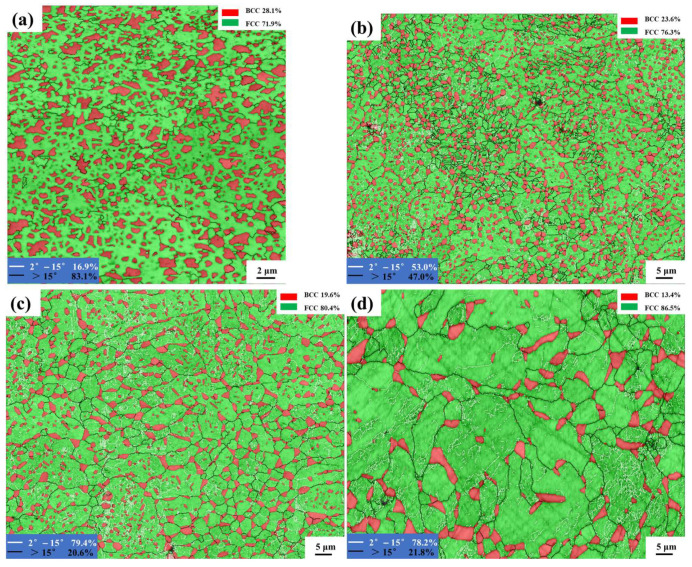
Phase distribution of the as-printed HEA (**a**) and heat-treated samples: 1000W-1 (**b**), 1100W-1 (**c**), 1200W-1 (**d**).

**Figure 5 micromachines-15-00471-f005:**
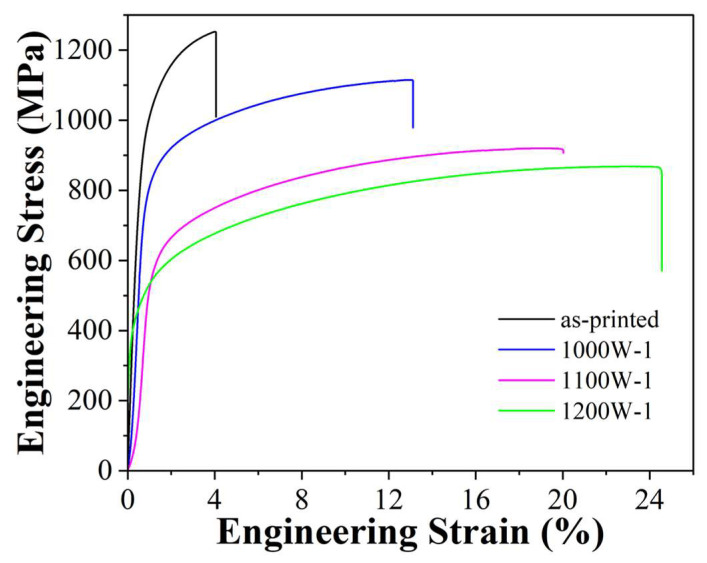
Tensile stress-strain curves at room temperature of as-printed and heat-treated HEA samples with different parameters.

**Figure 6 micromachines-15-00471-f006:**
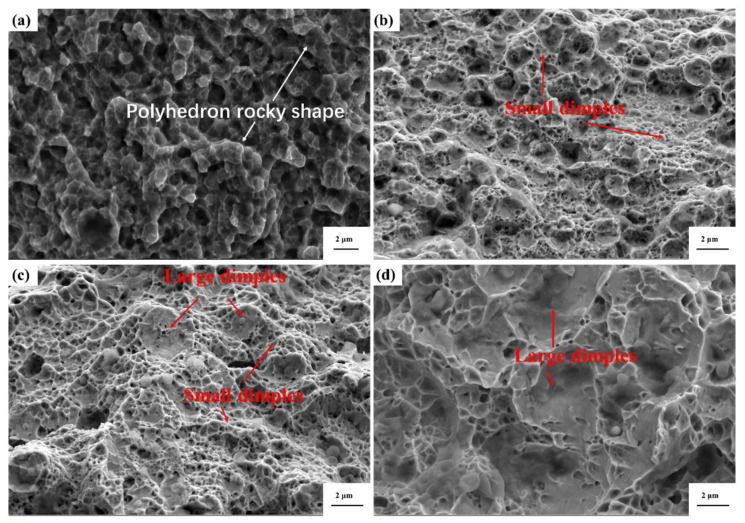
Fracture morphology of the as-printed HEA (**a**) and heat-treated samples: 1000W-1 (**b**), 1100W-1 (**c**), 1200W-1 (**d**).

**Figure 7 micromachines-15-00471-f007:**
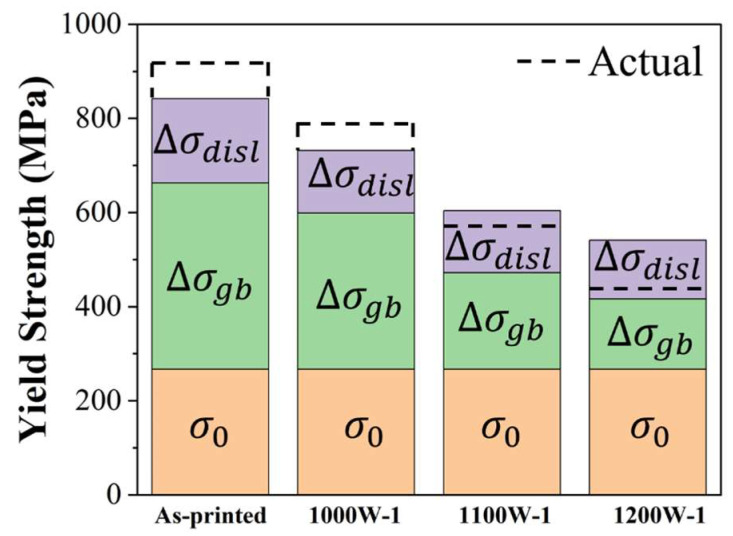
Strength contributions from different strengthening mechanisms of as-printed; 1000W-1; 1100W-1; and 1200W-1 samples.

**Figure 8 micromachines-15-00471-f008:**
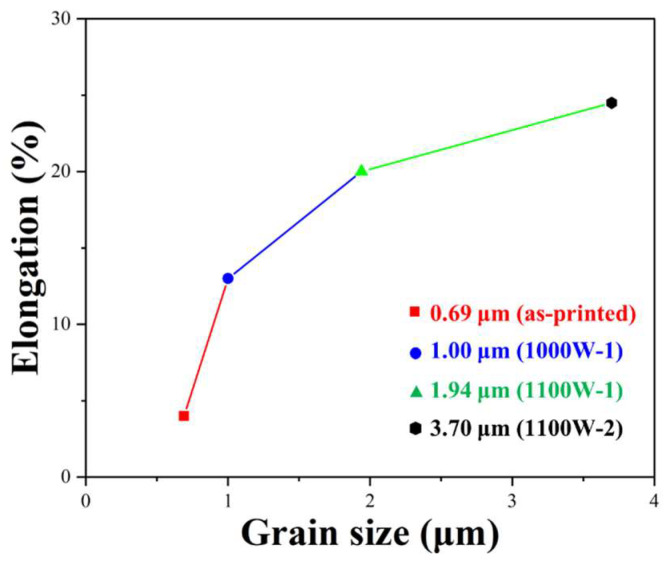
The relationship between elongation and grain size.

**Table 1 micromachines-15-00471-t001:** Heat-treated parameters after LB-PBF.

Sample	Temperature	Heat Preservation Time	Cooling Method
1000W-1	1000 °C	1 h	Water-cooled
1100W-1	1100 °C	1 h	Water-cooled
1200W-1	1200 °C	1 h	Water-cooled

**Table 2 micromachines-15-00471-t002:** The average phase sizes and their content and HAGB content.

Sample	$\mu_{BCC}$	$d_{FCC}$	$d_{BCC}$	$\mu_{HAGBs}$
As-printed	28.1%	1.20 μm	0.56 μm	83.1%
1000W-1	23.6%	1.33 μm	0.82 μm	47.0%
1100W-1	19.6%	4.33 μm	1.29 μm	20.6%
1200W-1	13.4%	5.8 μm	2.32 μm	21.8%

**Table 3 micromachines-15-00471-t003:** Mechanical properties of tensile tests.

Sample	Yield Strength (MPa)	Tensile Strength (MPa)	Elongation (%)
As-printed	913	1239	4.0
1000W-1	790	1115	13.0
1100W-1	571	920	20.0
1200W-1	430	868	24.5

**Table 4 micromachines-15-00471-t004:** The dislocation density of FCC and BCC phases in each sample and the calculated $\Delta \sigma_{disl}$.

Sample	$\rho_{FCC}$ (×10^14^ m^−2^)	$\rho_{BCC}$ (×10^14^ m^−2^)	$\Delta \sigma_{disl}$ (MPa)
As-printed	215	306	179
1000W-1	137	86	133
1100W-1	130	77	132
1200W-1	111	67	125

## Data Availability

The data that support the findings of this study are available from the corresponding author upon reasonable request.

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
