# Peer review of "Effects of Heat Treatment on the Microstructure and Mechanical Properties of a Dual-Phase High-Entropy Alloy Fabricated via Laser Beam Power Bed Fusion"

_micromachines, 2024, doi:10.3390/mi15040471_

Round 1

Reviewer 1 Report

Comments and Suggestions for Authors

The manuscript is devoted to an interesting issue - heat treatment of products produced by additive technologies.

The research was carried out at a fairly high level, but there are a number of comments:

1) Why was the holding time for heat treatment chosen as 1 hour? Does this depend on the thickness of the resulting product?

2) Justify the choice of heat treatment temperature.

3) In what atmosphere did the heat treatment take place? In air or argon?

4) Fig. 1. If possible, it is necessary to numerically show changes in the parameters of the fcc crystal lattice. Why is this not observed for a bcc lattice?

5) Fig. 1. A decrease in the theta angle for an fcc solid solution indicates a change in the solubility of elements in this solid solution. It is necessary to characterize this fact in more detail.

Author Response

Dear Prof. Reviewer #1, Editor, and Editor in chief,

Thank you very much for giving us an opportunity to revise our manuscript. We appreciate the editor and reviewers very much for their constructive comments and suggestions on our manuscript (micromachines-2884807) entitled “Journal name: Micromachines”.

We have carefully studied the comments of the Reviewer #1. In this response letter, our indications pertaining to the revision are following each comment of Reviewer #1.

Accordingly, we have uploaded a copy of the original manuscript with all the changes highlighted by using Red color. We hope that the revised manuscript is accepted for publication in the Journal of Micromachines.

Sincerely,

Dr. Xiaojun Tan

Dr. Zihong Wang

Mr. Haitao Chen

Prof. Xuyun Peng

Dr. Wei Zhang

Prof. Haibing Xiao*

Dr. Zhongmin Liu

Dr.Yu Hu

Dr. Liang Guo*  

Prof. Qingmao Zhang

March 19, 2024

Reviewer 2 Report

Comments and Suggestions for Authors

This study compareed the microstructure and mechanical properties of the as-printed sample with those cooled in water after undergoing heat treatment at temperatures ranging from 1000 to 1200 ℃. The effect of microstructure after heat treatment on the mechanical properties of dual phase high entropy alloys was discussed in this paper. However, it is recommended to supplement the composition and microstructure analysis of FCC/BCC phases.

Author Response

Dear Prof. Reviewer #2, Editor, and Editor in chief,

Thank you very much for giving us an opportunity to revise our manuscript. We appreciate the editor and reviewers very much for their constructive comments and suggestions on our manuscript (micromachines-2884807) entitled “Journal name: Micromachines”.

We have carefully studied the comments of the Reviewer #2. In this response letter, our indications pertaining to the revision are following each comment of Reviewer #2.

Accordingly, we have uploaded a copy of the original manuscript with all the changes highlighted by using Red color. We hope that the revised manuscript is accepted for publication in the Journal of Micromachines.

Sincerely,

Dr. Xiaojun Tan

Dr. Zihong Wang

Mr. Haitao Chen

Prof. Xuyun Peng

Dr. Wei Zhang

Prof. Haibing Xiao*

Dr. Zhongmin Liu

Dr.Yu Hu

Dr. Liang Guo*  

Prof. Qingmao Zhang

March 19, 2024

Reviewer 3 Report

Comments and Suggestions for Authors

Effects of Heat Treatment on the Microstructure and Mechanical Properties of a Dual-Phase High-Entropy Alloy Fabricated via Selective Laser Melting

Reviewer comments  – March 2024

I am including the article as an annotated PDF, which contains many comments, suggestions and questions to the authors. I am including only general comments and suggestions here.

     1  Overview of the topic

         1.1  Relevance

The work is relevant, because HEAs seem to present interesting potential for fabrication via additive manufacturing. The methods and results are mostly correct. However, the topic seems to have been covered somewhat extensively in the literature already, so I struggle to identify a significant contribution that this paper brings. 

     2  Analysis of the paper

         2.1  Abstract

             2.1.1  The abstract is mostly clear, as it presents a summary of motivations, methodologies and main conclusions. It mirrors the contents of the work without significant issues.

         2.2  Introduction

             2.2.1  The introduction provides an overview of the issue and a comprehensive review of the literature. After reading the section, the contribution of this work to the existing body of knowledge is not fully clear, mainly because there are works that study similar alloys with HTs at several temperatures; and although the authors claim that those studies are limited “because too less temperatures were chosen, and the heat treatment efficiency was not taken into consideration.” one can identify several studies with more temperature levels (e.g. ref [11] studies 5 temperatures). The authors also only study 1 HT time, with a poor justification, in my opinion (see PDF comment). I would recommend clarifying these issues.

             2.2.2  Suggestions / corrections: see PDF, besides the comments above.

         2.3  Materials and methods

             2.3.1  Sample preparation: I would recommend clarifying the amount of specimens manufactured as well as the number of LB-PBF batches. Also see the PDF for more comments.

             2.3.2  Phase and Microstructural Characterization: no major issues. the only question that arose for me was about the choice of the XY plane for observation, since the XZ should allow the observation of the columnar grain structure after manufacturing. See the PDF for details.

         2.4  Results 

             2.4.1  This chapter presents 2 sections, phase and microstructure, and mechanical properties. Overall I saw no major issues. Please see the PDF for some comments / suggestions.

         2.5  Discussion

             2.5.1  The discussion chapter starts with the microstructure evolution. I saw no major issues here, as the previous observations seem well explained through existing literature. Then there is the section on the study of the strengthening mechanisms. Two mechanisms are presented: grain size and dislocation density. Then, these factors are related to both the evolution of strength and ductility of the samples upon HT. Overall the approach seems correct, and the limitations of the results are also readily pointed out by the authors. However, I would recommend addressing my comment on line 340 (see PDF), since I believe it would improve the quality of the modeling of the yield strength presented.

             2.5.2  As a final suggestion for this chapter, in my opinion there is only 1 factor missing in the analysis: texture. This factor should also be discussed, in light of the texture evolution observed during heat treatments.

         2.6  Conclusions

             2.6.1  The conclusions chapter is a summary of the main findings, and those are correct. As suggestions for improvement, I would mention: the content pertaining to the modeling of the yield strength should also be mentioned here, since it is a significant part of the work; and the conclusion about the combination of strength / ductility (line 381) is only mentioned in the abstract and here. I think it should be mentioned in the discussion chapter as well. Moreover, I suggest using a quantifiable metric to define “a better combination of yield strength and ductility”, for instance the amount of plastic work, calculated via the true stress – true strain curves.

     3  Final remarks

       The present work is relevant. There are important questions that arose when reading the document, and that I would like to see clarified. The main issue pertains to the added value to the existing body of knowledge, which was not very clear in my opinion.

       As suggestions for improvement, and besides addressing the comments on the PDF already mentioned, the references should be thoroughly verified as well, as they seem incorrectly formed in some cases.

Author Response

Dear Prof. Reviewer #3, Editor, and Editor in chief,

Thank you very much for giving us an opportunity to revise our manuscript. We appreciate the editor and reviewers very much for their constructive comments and suggestions on our manuscript (micromachines-2884807) entitled “Journal name: Micromachines”.

We have carefully studied the comments of the Reviewer #3. In this response letter, our indications pertaining to the revision are following each comment of Reviewer #3.

Accordingly, we have uploaded a copy of the original manuscript with all the changes highlighted by using Red color. We hope that the revised manuscript is accepted for publication in the Journal of Micromachines.

Sincerely,

Dr. Xiaojun Tan

Dr. Zihong Wang

Mr. Haitao Chen

Prof. Xuyun Peng

Dr. Wei Zhang

Prof. Haibing Xiao*

Dr. Zhongmin Liu

Dr.Yu Hu

Dr. Liang Guo*  

Prof. Qingmao Zhang

March 19, 2024

Round 2

Reviewer 1 Report

Comments and Suggestions for Authors

The manuscript can be accepted for publication.

Author Response

Thank you for your excellent review

Reviewer 2 Report

Comments and Suggestions for Authors

The author provided explanation of the phase composition of high entropy alloys. The revised manuscript can be accepted for publication in the Journal of Micromachines.

Author Response

Thank you for your excellent review

Reviewer 3 Report

Comments and Suggestions for Authors

This is the review of the reply from the authors, to whom I thank for the attention. I used the author's response letter (attached) and added my comments on their replies. Many other small corrections in the text were done, as highlighted by the authors, to whom I thank for the attention. Only a few details remain to review, in my opinion, but those are clarified in the attached document.

Author Response

Dear Prof. Reviewer #3, Editor, and Editor in chief,

Thank you very much for giving us an opportunity to revise our manuscript. We appreciate the editor and reviewers very much for their constructive comments and suggestions on our manuscript (micromachines-2884807) entitled “Journal name: Micromachines”.

We have carefully studied the comments of the Reviewer #3. In this response letter, our indications pertaining to the revision are following each comment of Reviewer #3.

Accordingly, we have uploaded a copy of the original manuscript with all the changes highlighted by using red color. We hope the revised manuscript is accepted for publication in the Journal of Micromachines.

Sincerely,

Dr. Xiaojun Tan

Dr. Zihong Wang

Mr. Haitao Chen

Prof. Xuyun Peng

Dr. Wei Zhang

Prof. Haibing Xiao*

Dr. Zhongmin Liu

Dr.Yu Hu

Dr. Liang Guo*  

Prof. Qingmao Zhang

March 28, 2024
